# Long-Term Survival of Patients with Unresectable Hepatocellular Carcinoma Treated with Lenvatinib in Real-World Clinical Practice

**DOI:** 10.3390/cancers17030479

**Published:** 2025-02-01

**Authors:** Junji Furuse, Namiki Izumi, Kenta Motomura, Yoshitaka Inaba, Yoshio Katamura, Yasuteru Kondo, Kazuhisa Yabushita, Toshiyuki Matsuoka, Katsuaki Motoyoshi, Masatoshi Kudo

**Affiliations:** 1Department of Gastroenterology, Kanagawa Cancer Center, Yokohama 241-8515, Japan; 2Department of Gastroenterology and Hepatology, Musashino Red Cross Hospital, Musashino 180-8610, Japan; nizumi627@gmail.com; 3Department of Hepatology, Aso Iizuka Hospital, Iizuka 820-8505, Japan; kmotomurah2@aih-net.com; 4Department of Diagnostic and Interventional Radiology, Aichi Cancer Center Hospital, Nagoya 464-8681, Japan; 105824@aichi-cc.jp; 5Department of Gastroenterology, Onomichi General Hospital, Onomichi 722-8508, Japan; katamuray@yahoo.co.jp; 6Department of Hepatology, Sendai Tokushukai Hospital, Sendai 981-3116, Japan; yasuteru@ebony.plala.or.jp; 7Department of Internal Medicine, Fukuyama City Hospital, Fukuyama 721-8511, Japan; azuki.yabushita2002@gmail.com; 8Eisai Co., Ltd., Bunkyo-ku 112-8088, Japan; t-matsuoka@hhc.eisai.co.jp (T.M.); k-motoyoshi@hhc.eisai.co.jp (K.M.); 9Department of Gastroenterology and Hepatology, Kindai University Faculty of Medicine, Osakasayama 589-8511, Japan; m-kudo@med.kindai.ac.jp

**Keywords:** hepatocellular carcinoma, lenvatinib, overall survival, real-world practice

## Abstract

In clinical trials, lenvatinib has shown efficacy in terms of survival in patients with unresectable hepatocellular carcinoma. A large-scale prospective study is needed to confirm long-term survival after lenvatinib treatment in clinical practice. Therefore, we conducted a prospective, observational extension of a large-scale observational post-marketing study of lenvatinib, following patients for up to 3 years after lenvatinib treatment in clinical practice. We aimed to evaluate the long-term survival and associated factors. In 703 patients, the median overall survival (duration from the first lenvatinib dose to death from any cause) was 16.6 months. Overall survival was associated with invasion to the bile ducts and portal vein, and intra- and extra-hepatic lesions, the Child–Pugh class, and mALBI grade. These results demonstrated prolonged survival after lenvatinib treatment. More advanced-stage tumors and worse hepatic function have been suggested as overall-survival-associated factors, which is consistent with previous reports.

## 1. Introduction

Liver cancer is one of the most fatal types of cancer with a poor survival prognosis, ranking as the third most common cause of cancer-related deaths worldwide [1]. In Japan, although mortality is declining, it still ranks as the fifth leading cause of cancer-related deaths [2], with an annual incidence of 34,275 cases and 21,876 deaths due to liver cancer, estimated from 2020 to 2024 [3]. Approximately 80% of liver cancer cases are due to hepatocellular carcinoma (HCC) worldwide [4] and 90% in Japan [5]. The guidelines issued by The Japan Society of Hepatology recommend that combined immunotherapies, such as atezolizumab plus bevacizumab and durvalumab plus tremelimumab, be considered as a first-line systemic therapy for advanced HCC [6]. Sorafenib and lenvatinib are recommended for patients who are unsuitable candidates for combination treatment [7].

Lenvatinib is an oral tyrosine kinase inhibitor (TKI) that targets fibroblast growth factor (FGF) receptors 1–4, VEGF receptor 1–3, platelet-derived growth factor receptor α, the rearranged during transfection (RET) oncogene, and KIT [8,9,10]. Lenvatinib showed an antitumor effect against HCC cells by FGF signaling inhibition in a preclinical model [11]. An international phase 3 clinical trial (REFLECT trial) demonstrated that lenvatinib was non-inferior to sorafenib for overall survival (OS) in patients with unresectable hepatocellular carcinoma (uHCC) [12]. In Japan, lenvatinib was approved in March 2018 for the indication of uHCC.

The efficacy and safety profiles demonstrated in the clinical trial were derived from a limited patient population selected using an extensive list of inclusion/exclusion criteria [12,13,14], and these profiles may not be directly extrapolated to the overall population in actual clinical practice. Therefore, its evaluation in routine clinical practice is warranted. We previously conducted a prospective observational post-marketing study of lenvatinib and evaluated its safety and effectiveness in a clinical practice setting of patients with uHCC in Japan [15]. During the 1-year observation period, lenvatinib treatment was generally tolerated and treatment responses were clinically meaningful. We also reported a median OS of 498.0 days, 16.3 months [15]. However, the 1-year follow-up period was shorter than the median OS reported in the REFLECT trial (13.6 months) [12]. Therefore, it was considered too short to evaluate the survival of patients with uHCC after starting lenvatinib treatment. Moreover, since the approval of lenvatinib for the treatment of uHCC, other observational studies have explored the OS among patients with advanced HCC treated with lenvatinib in daily clinical practice [16,17,18,19]; however, these studies were conducted retrospectively with a relatively small sample size and short observation periods.

Therefore, we conducted a prospective, multicenter, post-marketing observational extension study of lenvatinib to sequentially follow patients with uHCC included in a preceding post-marketing study for up to 3 years to evaluate survival and factors associated with prognosis.

## 2. Materials and Methods

### 2.1. Study Design

This is a prospective, multicenter, observational extension study (referred to as the “510 study”, hereafter; ClinicalTrials. Gov Registration ID: NCT04008082) of the previous post-marketing study of lenvatinib (Lenvima^®^; Eisai Co., Ltd., Tokyo, Japan) (referred to as the “504 study”, hereafter) [15]. By extending the 1-year observation period of the 504 study by up to 2 years, we evaluated the long-term survival of patients treated with lenvatinib in terms of OS for a total of 3 years after the initiation of lenvatinib treatment.

The details of the 504 study were described by Furuse et al. [15]. In summary, the 504 study used the same design and was conducted as a prospective, multicenter, observational study (ClinicalTrials. Gov Registration ID: NCT03663114). From 137 institutions, the study enrolled 713 patients with uHCC who had never received lenvatinib and provided informed consent for study participation between July 2018 and January 2019. Lenvatinib was administered to the enrolled patients and followed up for 1 year after the first lenvatinib dose to evaluate its safety and effectiveness. Before the end of the 1-year observation period of the 504 study, eligible patients (described in the following section) were registered for a 2-year extension, the 510 study, from April 1 2019 to March 31 2020. Regardless of the treatment status (completion or discontinuation within 3 years), patients were followed up until the total observation period of 3 years after the first lenvatinib dose had been completed. Those who were lost to follow-up because of death or hospital transfer were followed up until the time of loss to follow-up.

The 510 study was conducted in accordance with the Declaration of Helsinki, Pharmaceutical Affairs Law, and Good Post-Marketing Study Practice (GPSP) in Japan. The study was approved by the ethics committee or institutional review board at each institute prior to initiation. Using data obtained from the 504 study period, we conducted a post hoc analysis to explore the profiles of patients who could be likely candidates for long-term lenvatinib treatment and the results have been reported elsewhere [20].

### 2.2. Patients and Treatment

The patients who were included in the 504 study and who provided new consent for participation in the 510 study, irrespective of their lenvatinib treatment continuation status, were eligible for the 510 study and were registered in a central registration method via an electronic data capture (EDC) system.

Lenvatinib was orally administered once daily. The standard lenvatinib dose, calculated on the basis of the patients’ body weights, was 12 mg/day for ≥60 kg and 8 mg/day for <60 kg. It was permissible to reduce the dose at the discretion of the treating physician according to the patient’s circumstances.

### 2.3. Data Collection

We used data collected using case report forms (CRFs) via the EDC system during the entire 3-year period, including demographic and clinical characteristics at the baseline of the observation period of the 504 study, the history of HCC treatment, and information regarding lenvatinib treatment (e.g., dosage, duration, and dose modifications), and survival outcome (alive or dead, and cause of death). Post-lenvatinib treatment for HCC, which had not been defined as an assessment item in the 504 study, was collected for the 510 study. Data during the 510 study periods were collected from two CRFs, one for the second year and the other for the third year after the first lenvatinib dose. For patients who were registered in the 510 study < 1 year after the first lenvatinib dose, the data on the post-lenvatinib treatment during the first two years were collected on the second-year CRF of the 510 study. Safety data were not collected during the 510 study and are not included in the present analysis or this paper. Safety during the 504 study was reported by Furuse et al. [15].

### 2.4. Assessment and Definition

Survival was assessed in terms of OS, which was defined as the duration from the first lenvatinib dose to death from any cause. In the case of a loss of follow-up, patients were censored on the last date on which they were alive.

The baseline liver function was assessed using the Child–Pugh classification and modified albumin–bilirubin (mALBI) grades. Based on the albumin–bilirubin (ALBI) score calculated by the formula (log_10_ bilirubin [μmol/L] × 0.66) + (albumin [g/L] × −0.085) [21], the mALBI grade was defined as follows: mALBI Grade 1: an ALBI score of ≤ −2.60; Grade 2a: > −2.60 to < −2.27; Grade 2b: ≥ −2.27 to ≤ −1.39; and Grade 3: > −1.39 [22]. As a baseline renal function measure, the estimated glomerular filtration rate (eGFR) was calculated using the equation: eGFR (mL/min/1.73 m^2^) = 194 × serum creatinine (mg/dL)^−1.094^ × age^−0.287^ (× 0.739, if female).

To assess the treatment status, the relative dose intensity (RDI) during the entire treatment period was calculated as the ratio of the actual total dosage administered to the standard dose (12 mg for patients weighing ≥ 60 kg or 8 mg for those weighing < 60 kg) multiplied by the treatment duration (days).

### 2.5. Statistical Analysis

The results presented in this paper include data obtained from the 504 and 510 studies. The present analysis was conducted on patients included in the safety and effectiveness analysis of the 504 study [15]. The baseline characteristics and lenvatinib treatment status are summarized descriptively. The median (95% confidence interval [CI]) OS in months was estimated using the Kaplan–Meier method. As a post hoc analysis, OS was also calculated for the subgroup of patients who would be eligible for the REFLECT trial, according to the following trial inclusion criteria: an Eastern Cooperative Oncology Group performance status (ECOG PS) of 0 or 1, Barcelona Clinic Liver Cancer (BCLC) stage B or C, Child–Pugh class A, and exclusion criteria of a history of chemotherapy, bile duct invasion, and main portal vein invasion [12].

To assess the factors associated with OS, first, the hazard ratios (HR) and 95% CI were estimated for each factor in the univariate analysis. Subsequently, multivariate Cox regression analysis was performed. The following baseline factors were entered as explanatory variables using stepwise methods at a significance criterion of *p* < 0.05: sex, age, body mass index (BMI), ECOG PS, bile duct invasion, portal vein invasion, maximum tumor size, number of intrahepatic lesions, extrahepatic lesions, history of transcatheter arterial chemoembolization (TACE), history of hepatic arterial infusion chemotherapy (HAIC), mALBI grade, eGFR, alpha-fetoprotein (AFP) level, and an initial dose of lenvatinib. In addition to these factors, we added the following two factors post hoc: a history of chemotherapy and a Child–Pugh class, which were included in the REFLECT trial eligibility criteria. All factors with *p* < 0.05 were considered statistically significant. The OS of the subgroups stratified by the identified OS-associated factors was visualized using Kaplan–Meier curves.

We also described the baseline characteristics and treatment status in subgroups stratified by the initial lenvatinib dose against the standard dose: the standard dosage group and reduced dosage group. As only a few patients (*n* = 6) were administered higher-than-standard doses, they were not included in the analysis of the initial dose subgroups reported in this manuscript.

For the analyses, SAS (version 9.4; SAS Institute, Inc., Cary, NC, USA) was used.

## 3. Results

### 3.1. Case Composition

Of the 713 patients registered in the 504 study, CRFs were collected from 708 patients, of whom five were excluded (two for protocol violation, one for lenvatinib administration for a non-indicated disease, one for no lenvatinib administration, and one for an uncertain AE status). The remaining 703 patients were included in the safety and effectiveness analysis of the 504 study, and the present analysis was conducted on these patients. The analysis set consisted of 410 patients whose data for both the 504 and 510 studies were available and 293 patients whose data for only the 504 study were available (Figure 1). In other words, 410 patients were included in the extension, the 510 study, whereas 293 patients were not included in the 510 study, and their follow-ups ended in the 504 study (112 patients: treating institutions were not under the contract for the 510 study, and 181 patients did not consent to participate in the 510 study). Of these 293 patients, 190 died and 103 were censored during the 504 study period. Of the 410 patients who were followed-up in the 510 study, 255 died and 155 were censored. The median (minimum, maximum) follow-up period for the 703 patients was 12.5 months (0.1, 44.8), with 445 deaths.

### 3.2. Baseline Demographic and Clinical Characteristics

Table 1 summarizes the baseline characteristics of the 703 patients included in the analysis. As previously reported [15], ECOG PS was 0 for 75.8% and ≥ 2 for 2.3% of patients. Seventy patients (10%) had bile duct invasion; 165 patients (23.5%) had portal vein invasion, including 3.1% (22/703 patients) having portal invasion, grade Vp4; and 60.2% and 32.6% had ≥ 4 intrahepatic lesions and extrahepatic lesions, respectively. Most of the patients (88.8%) were Child–Pugh class A and 75 patients (10.7%) were class B/C (class B, 73 patients; class C, 2 patients). Regarding HCC treatment history, 19.5% had received chemotherapy and 18.6% had received TKI therapy.

### 3.3. Treatment Status with Lenvatinib

The treatment status of lenvatinib is summarized in Table 2. Lenvatinib treatment was initiated at the standard dose in 519 (73.8%) patients (standard dosage group) and at the reduced dose in 178 (25.3%) patients (reduced dosage group). In the reduced dosage group, 121 patients weighed ≥ 60 kg; 93 of these patients (76.9%) started the treatment at 8 mg and 28 patients (23.1%) at 4 mg, whereas the weight-based standard was 12 mg. All 57 patients weighing < 60 kg started treatment at 4 mg, which was lower than the weight-based standard of 8 mg. Among the overall patients included in the analysis, the median period of lenvatinib treatment, including interruption, was 186.0 days, with 50.5% of patients experiencing treatment interruption. During the treatment period, the lenvatinib dose was reduced in 63.2% of patients, the RDI was ≥80% in 30.2% of patients, and the median RDI was 60.17%.

### 3.4. Survival

Among the 703 patients included in the present analysis, 445 (63.3%) died during the total observational period of 3 years, with a median OS of 16.6 months (95% CI: 15.4, 18.5) (Figure 2A). Applying the REFLECT trial eligibility criteria, 364 patients were found to be eligible for the trial; in this subgroup, the median OS was 18.0 months (95% CI: 15.8, 21.3) (Figure 2B).

Table 3 shows the factors associated with OS, explored in the univariate and multivariate Cox regression analyses. Based on the multivariate analysis, the following factors were significantly associated with OS: ECOG PS ≥ 1 (reference: 0) (HR: 1.778, *p* < 0.001); the presence of bile duct invasion (reference: absence) (HR: 1.621, *p* = 0.007); the presence of portal vein invasion (reference: absence) (HR: 1.365, *p* = 0.019); ≥4 intrahepatic lesions (reference: <4) (HR: 1.437, *p* = 0.001); the presence of extrahepatic lesions (reference: absence) (HR: 1.357, *p* = 0.007); Child–Pugh class B/C (HR: 1.515, *p* = 0.021); a higher mALBI grade, i.e., grade 2a (HR: 1.331, *p* = 0.045) and grade ≥ 2b (reference: grade 1) (HR: 1.811, *p* < 0.001); and the AFP level ≥ 200 ng/mL (reference: < 200 ng/mL) (HR: 1.690, *p* < 0.001).

Figure 3 shows the Kaplan–Meier curves of OS for the subgroups stratified by these OS-associated factors: EOCG PS (Figure 3A), bile duct invasion (Figure 3B), portal vein invasion (Figure 3C), number of intrahepatic lesions (Figure 3D), extrahepatic lesions (Figure 3E), Child–Pugh class (Figure 3F), mALBI grade (Figure 3G), and AFP level (Figure 3H). Of the factors relative to the REFLECT exclusion criteria, the median OS (95% CI) for the subgroup with and without bile duct invasion was 10.0 months (8.7, 14.3) and 17.6 months (15.8, 20.0), respectively (Figure 3B); that of the subgroup with Child–Pugh class A and B/C was 18.0 months (16.1, 20.4) and 7.7 months (5.9, 9.3), respectively (Figure 3F); and that of the subgroup with mALBI grade 1, 2a, and ≥ 2b was 22.5 months (19.2, 25.9), 19.2 months (15.8, 24.0), and 10.5 months (9.8, 12.2), respectively (Figure 3G).

The figure displays the Kaplan–Meier curve and median (95% CI) of the OS for the subgroups stratified by the factors suggested to be associated with OS [Table 3].

### 3.5. Characteristics and OS of the Standard Dosage Group and Reduced Dosage Group

Compared to the standard dosage group (*n* = 519), the percentage of the following characteristics was greater in the reduced dosage group (*n* = 178) by ≥10%: age of ≥75 years (standard dosage group vs. reduced dosage group: 40.3% vs. 51.1%, respectively); body weight of ≥60 kg (49.9% vs. 68.0%); ECOG PS ≥ 1 (19.3% vs. 35.4%); Child–Pugh class B or C (7.3% vs. 20.8%); and mALBI Grade ≥ 2b (34.3% vs. 54.5%) (Table 1).

As stated earlier, some patients started treatment at doses lower than recommended. Calculating the percentage among the reduced dosage group (178 patients), the initial dose was set at 8 mg in 52.2% (93 patients) and 4 mg in 15.7% (28 patients) against the 12 mg standard dose, and at 4 mg against the 8 mg standard dose in 32.0% (57 patients). In the reduced dosage group, the median duration of treatment (154.0 days) and the exposure (113.5 days) was shorter, and RDI during the treatment period (45.41%) was lower than in the standard dosage group (duration of treatment: 200.0 days, exposure: 170.0 days, and RDI: 68.44%) (Table 2). The OS was estimated for the standard dosage and reduced dosage subgroups as a reference, 17.7 months (95% CI: 15.8, 20.0) and 13.4 months (95% CI: 11.1, 16.6), respectively, and the initial dose of lenvatinib was not identified as a factor associated with OS.

## 4. Discussion

To the best of our knowledge, this is the first large-scale and long-term observational post-marketing study to evaluate survival after patients with uHCC started lenvatinib treatment in a real-world clinical setting. As previously reported [15], the 703 patients treated with lenvatinib mostly consisted of males, showing a similar demographic profile with that in the analysis population of previous Japanese studies [17,18,19]. Some patients included in the analysis had a history of chemotherapy, bile duct or portal vein invasion, or Child–Pugh class B or C. The present results reflect the actual state of treatment and long-term survival among patients in daily clinical practice, including those with more advanced-stage tumors and worse liver function who would be excluded from the REFLECT trial. Even including such patients, the median OS estimated from the present analysis population was 16.6 months, which is similar to the median OS of 17.6 months in the Japanese subset [14], and longer than the median OS of 13.6 months in the global overall population of the REFLECT trial [12]. Limiting the analysis to the subgroup of patients who would be eligible for the REFLECT trial (ECOG PS 0 or 1, BCLC stage B or C, Child–Pugh class A, and without a history of chemotherapy, bile duct invasion, or main portal vein invasion), the estimated OS was 18.0 months.

Our OS estimation for the subgroup reflecting the REFLECT trial population yielded a longer OS than that of the whole “real-world” study population. In a retrospective study conducted on 205 patients under clinical practice in Germany and Austria, the median OS was 12.8 months among the whole study population and 15.6 months in the subgroup that satisfied the eligibility criteria of the REFLECT trial [23]. Smaller retrospective studies conducted in Korea also demonstrated a longer OS in the subgroup simulating the REFLECT trial population than in the overall study population derived from clinical practice [24,25]. The longer OS in the REFLECT trial subgroup is considered reasonable given the stricter eligibility criteria for the clinical trial compared to observational studies reflecting daily clinical practice.

Patient background characteristics demonstrated that patients encountered in daily clinical practice might have more impaired liver function than those in the REFLECT trial. As suggested by the shorter median OS in the subgroup of Child–Pugh class B/C (7.7 months) than in those of class A (18.0 months), patients with a poorer liver function may have a poorer prognosis. In the analysis set, two patients were classified as Child–Pugh class C: one patient had an OS of 4.8 months and the other patient had an OS of 19.3 months. The other previous studies consistently reported the longer OS of Child–Pugh class A than class B/C, for example, 19.7 months vs. 4.1 months, respectively, in a prospective study of 59 patients, including those with tumor invasion [26], and 21.0 months vs. 9.0 months, respectively, in a retrospective study of 343 patients with uHCC [18]. Furthermore, in a retrospective study of 155 patients with uHCC, the median OS was 7.7 months for all patients, whereas an analysis limited to those with Child–Pugh class A showed a longer OS of 12.5 months [27].

Cox regression analysis showed that ECOG PS, bile duct invasion, portal vein invasion, the number of intrahepatic and extrahepatic lesions, the Child–Pugh class, mALBI Grade, and AFP level were associated with the OS. These factors, which reflect an impaired hepatic function and advanced-stage tumors, are generally consistent with those found in previous retrospective studies of patients with uHCC treated with lenvatinib in Japanese real-world practices. A retrospective study using real-world data from 343 patients with uHCC treated with lenvatinib suggested that the OS was associated with the ECOG PS, mALBI grade, AFP level, major vascular invasion, and history of molecular-targeted therapy, the latter of which was not identified in our study [18]. In a study by Hiraoka et al., mALBI grade 2b or 3 was the sole factor related to the OS [16], and Ogushi et al. found that Child–Pugh class B was associated with a worse OS [17]. Amioka et al. also found that ALBI grade 2b and AFP ≥ 400 ng/mL were negative predictors of the OS, while they did not find a significant association between OS and extrahepatic involvement [28]. In addition to these Japanese studies, a study conducted in Germany and Austria demonstrated patients with ALBI grade 2/3, an AFP level of ≥200 ng/mL, as well as EOCG PS ≥ 2 and macrovascular invasion were more likely to have shorter survival [23]. The present findings and previous reports suggest that the baseline pre-treatment liver function and tumor status are important prognostic factors relevant to OS.

Lenvatinib treatment started at a reduced dose in 25.3% of the patients. Compared to the standard dosage subgroup, the greater percentage of patients in the reduced dosage subgroup had EOCG PS ≥ 1, Child–Pugh class B or C, and mALBI grade ≥ 2b. For some of these patients, lenvatinib treatment started from a dose lower than the standard dose recommended based on body weight, considering the EOCG PS and liver function. We also inferred that for some patients, it might be started from a low dose to minimize the risk of untoward events and adjusted according to the tolerability, treatment response, and clinical course of each patient. Cosma et al. reported in their retrospective study of 28 patients with advanced HCC at their tertiary center that they also adopted a similar dosing scheme, starting at a dose lower than the standard one, followed by individual adjustment [29]. In the present reduced dosage and standard dosage subgroups, the median OS was 13.4 months and 17.7 months, respectively, while the initial dosage was not found to be associated with OS. In line with these results, in a retrospective observational study conducted on 100 Japanese patients with uHCC of Child–Pugh class A, the OS of the patients who started with a standard dosage (*n* = 51) was numerically longer than that of the patients who started with a reduced dosage (*n* = 49) (23.67 months vs. 13.64 months); nevertheless, the initial dosage was not identified as an independent OS predictive factor [30]. A phase I study conducted on 20 patients with advanced HCC resistant to standard therapy identified the daily maximum tolerable dose of lenvatinib to be 12 mg for those with Child–Pugh class A HCC and 8 mg for those with class B [31]. As the liver is the main elimination pathway of lenvatinib [32], reduced hepatic clearance increases the plasma concentration of lenvatinib, rendering dose reduction a reasonable option for patients with HCC. Collectively, these data suggest that adjusting the dose not only for body weight, but also for the clinical characteristics and conditions of each patient may contribute to optimizing lenvatinib treatment.

We also analyzed the OS among a subgroup of patients who had participated in the 510 study (*n* = 410) stratified by post-lenvatinib treatment (Appendix A). The median OS (95% CI) was 20.3 months (17.0, 23.5) for patients without post-lenvatinib treatment and 26.1 months (23.7, 27.4) for patients with post-lenvatinib treatment, with a significant inter-group difference (*p* = 0.016, log-rank test). The further stratification of patients with post-lenvatinib treatment into specific treatment modalities showed that the median OS for those who underwent radiotherapy, chemotherapy, TACE, and HAIC was 25.0, 24.7, 24.3, and 19.2 months, respectively, and the median OS was not reached in those who underwent surgery or percutaneous radiofrequency ablation. Even though HCC was unresectable before lenvatinib treatment, after treatment with lenvatinib, which induces a potent antitumor effect, it is considered that the size of the lesions was reduced to enable patients to undergo resection surgery or radiofrequency ablation.

However, there are some limitations to consider when interpreting these results. First, some patients who participated only in the 504 study and did not participate in the extension (the 510 study) were censored during the 504 study period. Second, data on the post-lenvatinib treatment of HCC were collected from the 510 study. Therefore, in the additional analysis presented in Appendix A, there may have been a selection bias attributed to the inclusion of only patients who were followed-up for an extended period. Nevertheless, the present data, which were derived from a large sample size of patients followed-up prospectively, are expected to contribute to better decision making for the lenvatinib treatment of patients with uHCC and for patients with diverse clinical characteristics, including those who are not reflected in clinical trials.

## 5. Conclusions

The OS after lenvatinib treatment in clinical practice was long and comparable to that reported in the REFLECT trial. OS has been suggested to be associated with bile duct invasion, portal vein invasion, extrahepatic lesions, Child–Pugh classes, and the mALBI grade.

## Figures and Tables

**Figure 1 cancers-17-00479-f001:**
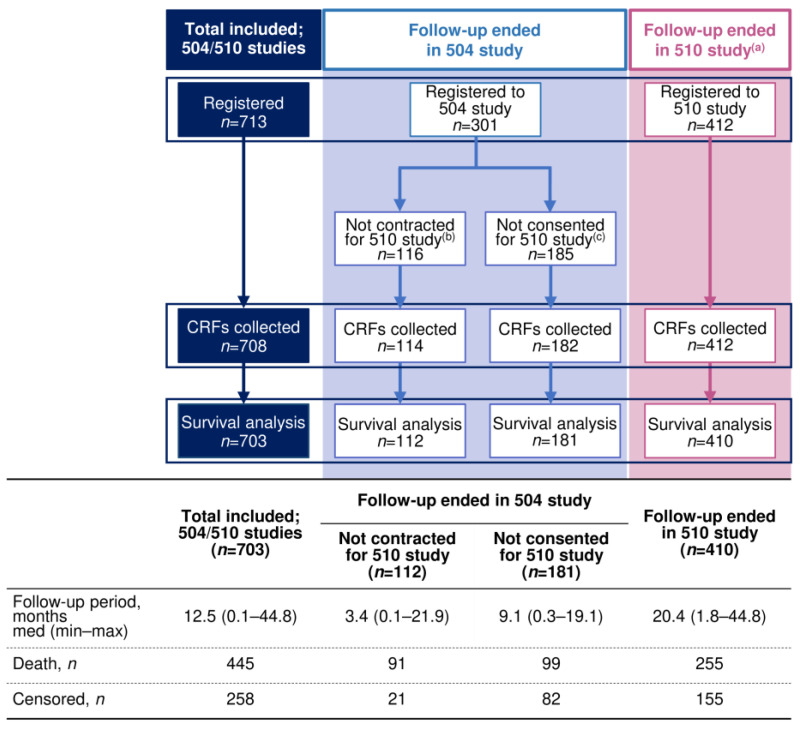
Flowchart of study registration for 504 and 510 studies. CRF, case report form; max, maximum; min, minimum. ^(a)^ The 510 study is an extension study of 504 study. ^(b)^ Patients were not included in the 510 study since their treating institutions were not under contract for the 510 study. ^(c)^ Patients were not included in the 510 study since they did not consent to participate in the 510 study while their treating institutions were under contract for the study.

**Figure 2 cancers-17-00479-f002:**
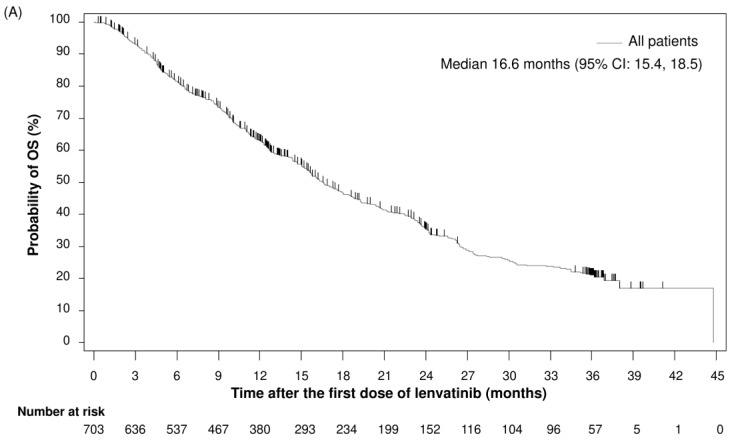
Kaplan–Meier estimates of OS in (**A**) all patients (*n* = 703) and (**B**) by the REFLECT trial eligibility criteria. Patients were classified into subgroups of whether they met (“Yes”) or did not meet (“No”) the eligibility criteria of the REFLECT trial: inclusion criteria of Eastern Cooperative Oncology Group performance status (ECOG PS) 0 or 1, Barcelona Clinic Liver Cancer (BCLC) stage B or C, Child–Pugh class A, and exclusion criteria of a history of chemotherapy, bile duct invasion, or main portal vein invasion. CI, confidence interval; OS, overall survival.

**Figure 3 cancers-17-00479-f003:**
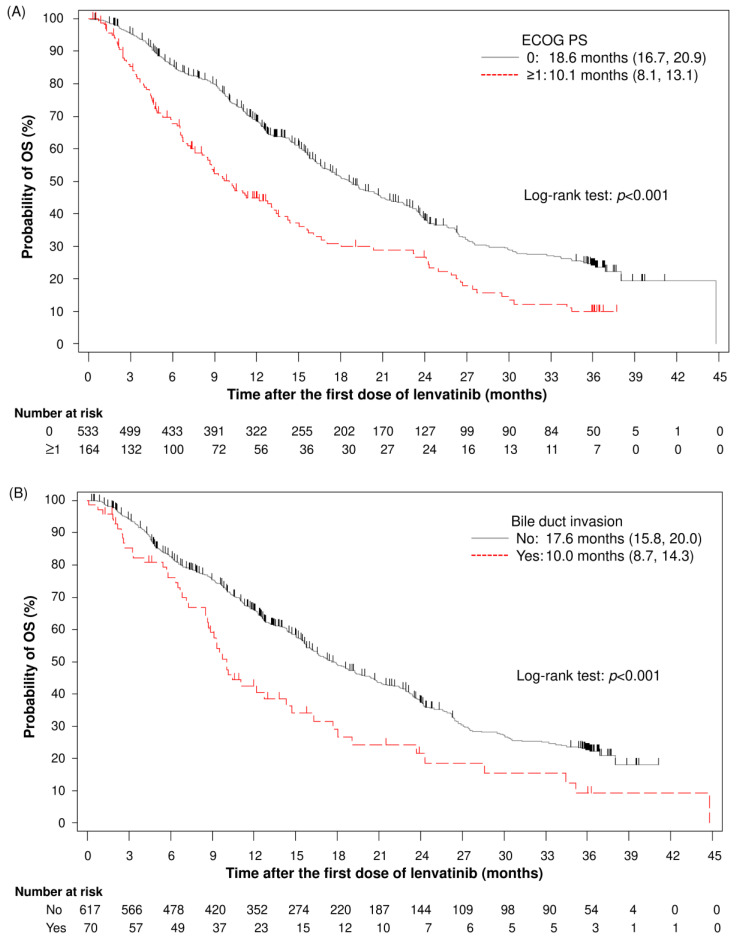
Kaplan–Meier estimates of OS by the factors associated with OS: EOCG PS (**A**), bile duct invasion (**B**), portal vein invasion (**C**), number of intrahepatic lesions (**D**), extrahepatic lesions (**E**), Child–Pugh class (**F**), mALBI grade (**G**), and AFP level (**H**). AFP, alpha-fetoprotein; ECOG PS, Eastern Cooperative Oncology Group performance status; mALBI, modified albumin–bilirubin; OS, overall survival.

**Table 1 cancers-17-00479-t001:** Demographic and clinical characteristics at baseline.

		Total ^(a)^ (*n* = 703)		Initial Dose of Lenvatinib			
				Standard Dosage ^(b)^ (*n* = 519)		Reduced Dosage ^(c)^ (*n* = 178)	
Gender, *n* (%)	Male	564	(80.2)	420	(80.9)	138	(77.5)
	Female	139	(19.8)	99	(19.1)	40	(22.5)
Age (years)	<65	117	(16.6)	91	(17.5)	25	(14.0)
	65–74	283	(40.3)	219	(42.2)	62	(34.8)
	≥75	303	(43.1)	209	(40.3)	91	(51.1)
	Median (min, max)	73.0 (25, 94)		72.0 (25, 94)		75.0 (39, 91)	
Body weight (kg), *n* (%)	<60	323	(45.9)	260	(50.1)	57	(32.0)
	≥60	380	(54.1)	259	(49.9)	121	(68.0)
BMI (kg/m^2^)	Median (min, max)	23.16 (13.9, 42.9)		22.89 (15.4, 42.9)		24.04 (13.9, 36.7)	
ECOG PS, *n* (%)	0	533	(75.8)	413	(79.6)	115	(64.6)
	1	148	(21.1)	91	(17.5)	56	(31.5)
	≥2	16	(2.3)	9	(1.7)	7	(3.9)
	Unknown	6	(0.9)	6	(1.2)	0	
BCLC stage, *n* (%)	Stage 0/A	59	(8.4)	40	(7.7)	19	(10.7)
	Stage B	291	(41.4)	220	(42.4)	69	(38.8)
	Stage C	332	(47.2)	243	(46.8)	85	(47.8)
	Stage D	11	(1.6)	9	(1.7)	2	(1.1)
	Unknown	10	(1.4)	7	(1.3)	3	(1.7)
Bile duct invasion, *n* (%)	No	617	(87.8)	459	(88.4)	153	(86.0)
	Yes	70	(10.0)	54	(10.4)	15	(8.4)
	Unknown	16	(2.3)	6	(1.2)	10	(5.6)
Portal vein invasion, *n* (%)	Vp0	520	(74.0)	400	(77.1)	117	(65.7)
	Vp1	31	(4.4)	19	(3.7)	10	(5.6)
	Vp2	52	(7.4)	36	(6.9)	16	(9.0)
	Vp3	60	(8.5)	42	(8.1)	18	(10.1)
	Vp4	22	(3.1)	16	(3.1)	6	(3.4)
	Unknown	18	(2.6)	6	(1.2)	11	(6.2)
Maximum tumor size (cm), *n* (%)	<3	277	(39.4)	210	(40.5)	62	(34.8)
	≥3 to <5	177	(25.2)	125	(24.1)	52	(29.2)
	≥5	229	(32.6)	169	(32.6)	59	(33.1)
	Unknown	20	(2.8)	15	(2.9)	5	(2.8)
Number of intrahepatic lesions, *n* (%)	<4	266	(37.8)	194	(37.4)	69	(38.8)
	≥4	423	(60.2)	315	(60.7)	105	(59.0)
	Unknown	14	(2.0)	10	(1.9)	4	(2.2)
Extrahepatic lesions, *n* (%)	No	450	(64.0)	340	(65.5)	106	(59.6)
	Yes	229	(32.6)	167	(32.2)	60	(33.7)
	Unknown	24	(3.4)	12	(2.3)	12	(6.7)
Child–Pugh class, *n* (%)	A	624	(88.8)	478	(92.1)	140	(78.7)
	B	73	(10.4)	37	(7.1)	36	(20.2)
	C	2	(0.3)	1	(0.2)	1	(0.6)
	Unknown	4	(0.6)	3	(0.6)	1	(0.6)
History of chemotherapy, *n* (%)	No	566	(80.5)	420	(80.9)	142	(79.8)
	Yes	137	(19.5)	99	(19.1)	36	(20.2)
History of TKI therapy, *n* (%)	No	572	(81.4)	426	(82.1)	142	(79.8)
	Yes	131	(18.6)	93	(17.9)	36	(20.2)
History of TACE (times), *n* (%)	0	190	(27.0)	157	(30.3)	31	(17.4)
	< 3	229	(32.6)	166	(32.0)	61	(34.3)
	≥ 3	272	(38.7)	189	(36.4)	81	(45.5)
	Unknown	12	(1.7)	7	(1.3)	5	(2.8)
History of HAIC, *n* (%)	No	629	(89.5)	474	(91.3)	149	(83.7)
	Yes	74	(10.5)	45	(8.7)	29	(16.3)
mALBI grade, *n* (%)	Grade 1	216	(30.7)	180	(34.7)	32	(18.0)
	Grade 2a	199	(28.3)	150	(28.9)	48	(27.0)
	≥Grade 2b	276	(39.3)	178	(34.3)	97	(54.5)
	Incalculable	12	(1.7)	11	(2.1)	1	(0.6)
AFP level (ng/mL), *n* (%)	<200	432	(61.5)	323	(62.2)	105	(59.0)
	≥200	240	(34.1)	169	(32.6)	69	(38.8)
	Unknown	31	(4.4)	27	(5.2)	4	(2.2)

^(a)^ Includes six patients who weighed < 60 kg but started treatment at 12 mg, which was higher than the standard dosage of 8 mg. ^(b)^ Patients weighing < 60 kg with an initial lenvatinib dose of 8 mg and patients weighing ≥ 60 kg with an initial dose of 12 mg. ^(c)^ Patients weighing < 60 kg with an initial lenvatinib dose of <8 mg and patients weighing ≥ 60 kg with an initial dose of < 12 mg. AFP, alpha-fetoprotein; BCLC stage, Barcelona Clinic Liver Cancer stage; BMI, Body mass index; ECOG PS, Eastern Cooperative Oncology Group performance status; HAIC, hepatic arterial infusion chemotherapy; mALBI, modified albumin–bilirubin; max, maximum; min, minimum; TACE, transcatheter arterial chemoembolization; TKI, tyrosine kinase inhibitor.

**Table 2 cancers-17-00479-t002:** Status of treatment with lenvatinib.

		Total ^(a)^ (*n* = 703)		Initial Dose of Lenvatinib			
				Standard Dosage ^(b)^ (*n* = 519)		Reduced Dosage ^(c)^ (*n* = 178)	
Initial dose (mg/day), *n* (%)	12	265	(37.7)	259	(49.9)	0	
	8	353	(50.2)	260	(50.1)	93	(52.2)
	4	85	(12.1)	0		85	(47.8)
Duration of treatment (days)	Median (min, max)	186.0 (2, 1099)		200.0 (2, 1099)		154.0 (2,1096)	
Duration of exposure (days)	Median (min, max)	153.0 (2, 1096)		170.0 (2, 1096)		113.5 (2, 1096)	
Dose reduction, *n* (%)	Yes	444	(63.2)	350	(67.4)	88	(49.4)
Time to first dose reduction (days)	Median (min, max)	42.0 (2, 1099)		37.0 (2, 1099)		65.5 (4, 709)	
Interruption, *n* (%)	Yes	355	(50.5)	264	(50.9)	89	(50.0)
Duration of interruption (days)	Median (min, max)	30.0 (1, 674)		26.5 (1, 674)		43.0 (1, 549)	
Relative dose intensity	≥80	212	(30.2)	205	(39.5)	4	(2.2)
	<80 to ≥60	141	(20.1)	102	(19.7)	36	(20.2)
	<60 to ≥40	188	(26.7)	129	(24.9)	59	(33.1)
	<40	162	(23.0)	83	(16.0)	79	(44.4)
	Median (min, max)	60.17 (5.9, 112.8)		68.44 (5.9, 100.0)		45.41 (8.8, 98.1)	

^(a)–(c)^ See Table 1. max, maximum; min, minimum.

**Table 3 cancers-17-00479-t003:** Univariate (A) and multivariate (B) analyses for factors associated with OS.

Variables	Categories	Univariate Analysis				Multivariate Analysis			
		*n*	Event	HR (95% Confidence)	*p* Value	*n*	Event	HR (95% Confidence)	*p* Value
Sex	Male	564	356	Reference					
	Female	139	89	1.066 (0.844, 1.347)	*p* = 0.589				
Age (years)	<65	117	76	Reference					
	≥65 <75	283	190	0.876 (0.671, 1.143)	*p* = 0.329				
	≥75	303	179	0.968 (0.740, 1.266)	*p* = 0.810				
BMI (kg/m^2^)	<23.2	351	226	Reference					
	≥23.2	351	218	0.869 (0.721, 1.047)	*p* = 0.139				
ECOG PS	0	533	326	Reference		444	277	Reference	
	≥1	164	115	1.850 (1.494, 2.291)	*p* < 0.001	137	98	1.778 (1.402, 2.254)	*p* < 0.001
Bile duct invasion	No	617	381	Reference		521	332	Reference	
	Yes	70	51	1.740 (1.294, 2.340)	*p* < 0.001	60	43	1.621 (1.139, 2.307)	*p* = 0.007
Portal vein invasion	No	520	318	Reference		450	278	Reference	
	Yes	165	118	1.703 (1.377, 2.108)	*p* < 0.001	131	97	1.365 (1.052, 1.770)	*p* = 0.019
Maximum tumor size (cm)	<3	277	163	Reference					
	≥3 to <5	177	111	1.258 (0.987, 1.602)	*p* = 0.063				
	≥5	229	163	1.636 (1.316, 2.034)	*p* < 0.001				
Number of intrahepatic lesions	<4	266	149	Reference		219	121	Reference	
	≥4	423	289	1.335 (1.095, 1.627)	*p* = 0.004	362	254	1.437 (1.149, 1.797)	*p* = 0.001
Extrahepatic lesions	No	450	284	Reference		389	252	Reference	
	Yes	229	148	1.271 (1.042, 1.552)	*p* = 0.018	192	123	1.357 (1.086, 1.695)	*p* = 0.007
History of chemotherapy	No	566	356	Reference					
	Yes	137	89	1.020 (0.807, 1.288)	*p* = 0.870				
History of TACE (times)	0	190	120	Reference					
	<3	229	132	0.813 (0.635, 1.042)	*p* = 0.101				
	≥3	272	185	1.040 (0.826, 1.308)	*p* = 0.738				
History of HAIC	No	629	398	Reference					
	Yes	74	47	1.068 (0.789, 1.445)	*p* = 0.669				
Child–Pugh class	A	624	388	Reference		520	330	Reference	
	B/C	75	56	2.892 (2.171, 3.854)	*p* < 0.001	61	45	1.515 (1.064, 2.157)	*p* = 0.021
mALBI Grade	Grade 1	216	114	Reference		185	102	Reference	
	Grade 2a	199	124	1.268 (0.983, 1.636)	*p* = 0.067	168	104	1.331 (1.005, 1.762)	*p* = 0.045
	≥Grade 2b	276	199	2.234 (1.771, 2.817)	*p* < 0.001	228	169	1.811 (1.389, 2.360)	*p* < 0.001
eGFR (mL/min)	≥45	615	396	Reference					
	<45	72	39	1.041 (0.749, 1.446)	*p* = 0.813				
AFP level (ng/mL)	<200	432	253	Reference		381	228	Reference	
	≥200	240	172	1.775 (1.460, 2.156)	*p* < 0.001	200	147	1.690 (1.363, 2.095)	*p* < 0.001
Initial dose of lenvatinib	Standard dosage ^(a)^	519	329	Reference					
	Reduced dosage ^(b)^	178	114	1.261 (1.018, 1.562)	*p* = 0.033				

Factors associated with OS were analyzed using Cox regression analysis. Variables were entered into the multivariate model with stepwise selection at a significance criterion of *p* < 0.05. ^(a),(b)^ See Table 1 for the standard dosage and reduced dosage. AFP, alpha-fetoprotein; BMI, Body mass index; ECOG PS, Eastern Cooperative Oncology Group performance status; eGFR, estimated glomerular filtration rate; HAIC, hepatic arterial infusion chemotherapy; HR, hazard ratio; mALBI, modified albumin–bilirubin; TACE, transcatheter arterial chemoembolization.

## Data Availability

Individual data were provided to Eisai under a contract between the participating institutions and Eisai. Due to restrictions by Eisai, the underlying data may not be made publicly available. However, data are available to interested and qualified researchers who meet the criteria for access to confidential data upon request to the Clinical Trial Disclosure site of Eisai. For further information, please access the following links: https://www.eisai.co.jp/innovation/research/clinical_trials/clinical/index.html (Japanese) (accessed on 15 November 2024); https://www.eisai.com/innovation/research/clinical_trials/clinical/index.html?_gl=1*801ekg*_ga*MjI5NDMwMjY3LjE2OTgzNjcwNzE.*_ga_X1FWS6YR87*MTY5ODM2NzA3MS4xLjAuMTY5ODM2NzA3OC41My4wLjA (English) (accessed on 15 November 2024).

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
