# Peer review of "Long-Term Survival of Patients with Unresectable Hepatocellular Carcinoma Treated with Lenvatinib in Real-World Clinical Practice"

_cancers, 2025, doi:10.3390/cancers17030479_

Round 1

Reviewer 1 Report

Comments and Suggestions for Authors

The paper presented by the authors has resumed their work with a small cohort and brilliant results on survival in patients affected by HCC treated with lenvatinib and have made it, on that track, a prospective multicenter with a large population that was absolutely informed about the proposed therapy. As in the previous work they have opted for the same inclusion and exclusion criteria, they have used the now known reference data such as Child-Pugh and known instrumental tests. The results are those of a large reference center that treats a high volume of patients and that in our clinic we have noted for gastric neoplasms but that is well suited to this research. (doi.org/10.3390/jcm12072708 to be cited in the bibliography). I mostly agree with the sentence "However, the current data, which were derived from a large sample of prospectively followed patients, should contribute to a better decision-making process for lenvatinib treatment of patients with uHCC for patients with different clinical characteristics, including those not reflected in clinical trials". Clearly the conclusions could only be those written by authors, invasion of the bile ducts or portal vessels obviously makes the difference together with the high Child-Pugh grade. Good English, good bibliography on which the entire paper rests The work is accompanied by excellent iconography

Author Response

Reviewer 1

The paper presented by the authors has resumed their work with a small cohort and brilliant results on survival in patients affected by HCC treated with lenvatinib and have made it, on that track, a prospective multicenter with a large population that was absolutely informed about the proposed therapy. As in the previous work they have opted for the same inclusion and exclusion criteria, they have used the now known reference data such as Child-Pugh and known instrumental tests. The results are those of a large reference center that treats a high volume of patients and that in our clinic we have noted for gastric neoplasms but that is well suited to this research. (doi.org/10.3390/jcm12072708 to be cited in the bibliography). I mostly agree with the sentence "However, the current data, which were derived from a large sample of prospectively followed patients, should contribute to a better decision-making process for lenvatinib treatment of patients with uHCC for patients with different clinical characteristics, including those not reflected in clinical trials". Clearly the conclusions could only be those written by authors, invasion of the bile ducts or portal vessels obviously makes the difference together with the high Child-Pugh grade. Good English, good bibliography on which the entire paper rests The work is accompanied by excellent iconography

Our response

Thank you for your valuable comments and positive feedback. We appreciate your understanding and are grateful for your valuable input.

Reviewer 2 Report

Comments and Suggestions for Authors

This is a good manuscript, however, some points have to be fixed before the decision.

1) The title is inappropriately long, making the manuscript misunderstood. The authors must revise it.

2)      I highly recommend the authors draw a graphical abstract at the end of the introduction.

3)      I just would like to know the novelty of this work. Perhaps post-marketing surveillance has been performed by the inventor company of Levantinib.

4)      In data collection, what kind of data has been collected? Only survival or death?

5)      Is there any relationship found between death and the age of the patient? It is good for the authors to explain it since older people have a higher risk of death whether is death due to cancer or age.

6)      The quality of the figures 2 and 3 are very poor. The text inside of them cannot be seen. They have to revise them.   

Reviewer 3 Report

Comments and Suggestions for Authors

This would be the first large-scale (703 patients included) and long-term (following patients for up to 3 years after lenvatinib treatment in clinical practice) observational, multicenter and prospective post-marketing study to evaluate survival after patients with unresectable hepatocellular carcinoma (uHCC) started lenvatinib treatment in a real-world clinical setting.

General comment:

The topic of this article is highly important and the results are valuable and important to share with a scientific community. The manuscript is well written and mainly methodologically sound. The limitations of the study are fairly presented. However, there are a few minor points that should be addressed in order to improve the manuscript.

Specific points:

Figure 2 consists of two panels; however, in the description of a Figure 2 (Figure 2 captions) only 'a)' exists and no 'b)'.

Furthermore, Figure 3 lacks the statistical analysis (log-rank test) and a p-value (comparison between two curves).

Table S1 says: 'a) Estimated by the Kaplan–Meier method'. It is not clear why the authors did not show the corresponding Kaplan-Meier curves.

Reviewer 4 Report

Comments and Suggestions for Authors

There are some comments.

1. It would be better to add an analysis based on the pathological features of hepatocellular carcinoma cases (e.g., histological subtype, histological grade) and underlying liver diseases.

2. It would be better to clarify terms such as "Not contracted for the 510 study" and "Not consented for the 510 study."

3. It would be better to review the layout and modify it to improve the readability of the tables.

Comments on the Quality of English Language

Please check English grammar and spelling.

Round 2

Reviewer 2 Report

Comments and Suggestions for Authors

The manuscript has been improved, however, 1) the authors did not provide a graphical abstract ( they wrote that they prepared it) at the end of the introduction and 2) the quality of the figures 2 and 3 is inferior, they must improve them. 

Reviewer 4 Report

Comments and Suggestions for Authors

The manuscript was well-revised.

Please add abbreviations below the Tables.

Comments on the Quality of English Language

Please check English grammar and spelling.

For example, mALBI Grade, grade1 -> grade 1

                                            grade2a -> grade 2a

                                             grade2b -> grade 2b

Round 3

Reviewer 2 Report

Comments and Suggestions for Authors

The authors answered my comments, and I recommend publishing. The only remaining issue is the quality of the images is still poor. They must be improved before publishing and ensure they are at least 300 DPI.